# Pipeline Parallelism Optimization with Deep Reinforcement Learning

## Abstract

It has been widely observed that larger neural networks perform better in many real-world applications. While this scaling trend affirms the need to train a giant model across multiple devices, it is challenging to partition a model with millions of parameters to run efficiently and effectively on various devices deployed in a cluster of accelerators, e.g., GPUs and TPUs. Recently, a novel approach to distributed training deep neural network (DNN) models has been proposed, pipeline parallelism. Compared with data parallelism, the existing works achieved a significant speed-up ratio even with a naive partition scheme.

This paper presents a deep reinforcement learning (DRL)-based pipeline parallelism framework, *DRL-PP*, that learns to optimize the pipeline schedule for training large DNN models across multiple accelerators. The core of *DRL-PP* is a DRL agent consisting of a graph encoder, describing the semantics of an operator in the computational graph, followed by a recurrent model partitioner and a pipeline scheduler that learns to partition and place operations on various GPU devices automatically. In particular, by generating placement in a recurrent way, *DRL-PP* can partition DNN models in a more flexible and balanced manner, which improves accelerator utilization and speeds up DNN training. We deployed and extensively evaluated *DRL-PP* on various benchmarks. Compared with the state-of-the-art, *DRL-PP* can speed up the distributed training of benchmark models up to $6.8\times$ and $1.3\times$ over data parallelism and PipeDream, respectively.

## 1 Introduction

With the growth of machine learning, today, deep neural networks (DNNs) have been widely used in many real-world applications, and DNN models are becoming exceedingly large. For example, most state-of-the-art image classification models and natural language processing models (Brown et al., 2020; Zhai et al., 2022) have billions of parameters and take days or even weeks to train to satisfactory accuracy. To address the increasing training overhead of DNN models, it is common to use a cluster of accelerators, e.g., GPUs or TPUs, to speed up the training process.

However, it is non-trivial to distribute the DNN training task over a cluster. Data parallelism and model parallelism are the two most popular distributed training methods researchers have studied for many years. Data parallelism (Krizhevsky et al., 2012), namely, splits the machine learning task along the data dimensions. It distributes input data across the accelerators and processes data concurrently. At the end of each round, it aggregates results from all workers and updates the models on all workers. Unlike data parallelism, which requires storage of all parameters by each worker, model parallelism (Dean et al., 2012) splits the task along the parameter dimensions. Thus each worker only holds part of the model's parameters and processes the corresponding part of the training task. While data parallelism usually achieves better efficiency, model parallelism is a must when training exceedingly large models, e.g., Bidirectional Encoder Representations from Transformers (BERT) (Devlin et al., 2019), that cannot fit into a single accelerator's memory.

A novel distributed training method has recently been proposed, pipeline parallelism (Huang et al., 2019), which combines the advantages of data parallelism and model parallelism. Existing works (Narayanan et al., 2019; 2021) further improve the partition scheme and pipeline execution, which greatly speed up DNN training. While the idea of distributing tasks in a pipelined fashion

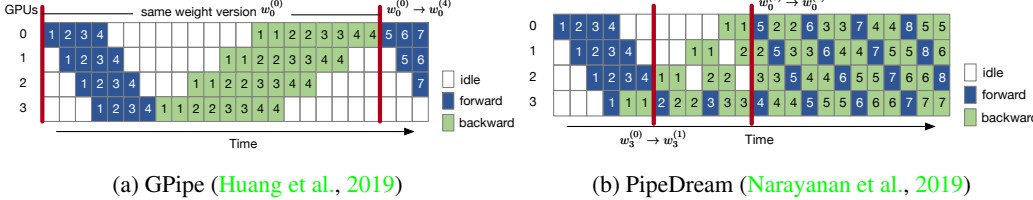

Figure 1: An illustration of pipeline schedule of existing pipeline parallelism frameworks.

is not novel, there are still many challenges when applying it to DNN training tasks, especially for complex DNN models.

In this paper, we propose a deep reinforcement learning (DRL) based pipeline parallelism framework, called *DRL-PP*, that learns to optimize pipeline placement for training large DNN models across multiple accelerators distributedly. The core of *DRL-PP* is a DRL agent consisting of three components, the graph encoder, recurrent partitioner, and pipeline scheduler. The graph encoder describes the semantics of operators in a computational graph by a graph convolutional network (GCN). The recurrent partitioner learns to partition the DNN models by traversing the computational graph node by node. With the pipeline scheduler, partitions are automatically assigned to several GPU devices with an optimal pipeline placement.

Unlike previous work, which treated the DNN model as a chain structure, the recurrent partitioner of the DRL-PP partitions DNN models without compressing their branches. Thus, the placements generated by *DRL-PP* are more flexible and balanced. Moreover, existing works assume that they can profile the runtime performance of DNN models on the clusters, which may not be true in practice. In contrast, *DRL-PP* uses deep reinforcement learning to automatically learn the statistics of the real-world environment by trial and error, which is more general and effective. As a result, *DRL-PP* can accelerate distributed training up to $6.8\times$ and $1.3\times$ over data parallelism and PipeDream respectively.

## 2 PRELIMINARIES AND RELATED WORKS

### 2.1 PIPELINE PARALLELISM

Due to the outstanding performance of large DNN models trained on a tremendous amount of data samples, today's DNN models are becoming exceedingly large. It takes days, even weeks, to train the DNN models to satisfactory accuracy. Thus there is an emerging need to speed up the training process with a cluster of accelerators, e.g., GPUs or TPUs. However, training a DNN model distributedly is non-trivial. Data parallelism and model parallelism are the two most popular distributed training methods researchers have studied for many years. Recently, a novel distributed training method has been proposed, pipeline parallelism, which combines the advantages of data parallelism and model parallelism. Pipeline parallelism has achieved significant speedup for DNN model training, especially in large scale distributed training.

GPipe (Huang et al., 2019) is the first work that proposes to use pipeline parallelism for distributed DNN training. First, it partitions the deep neural network across different accelerators. Then, as Figure 1a shown, GPipe splits a mini-batch of training samples into micro-batches and processes them in a pipeline fashion. Finally, after the gradients of all micro-batches have been computed, GPipe updates the model's parameters for all accelerators synchronously. Unlike data parallelism, the accelerators do not need to exchange gradients with each other since the gradients are computed on the accelerators that store the corresponding parameters and can be applied locally. The communication overhead of GPipe is the same as model parallelism, which is the intermediate results (activations and gradients) computed by each accelerator.

Beyond GPipe, PipeDream (Narayanan et al., 2019) pointed out that the "bubbles" (white blocks in Figure 1) in the pipeline can be further minimized. As Figure 1b shows, they use asynchronous weight updates, which compute gradients based on outdated weights. This approach eliminates the bubble caused by weight synchronization between accelerators. As a trade-off, PipeDream needs to

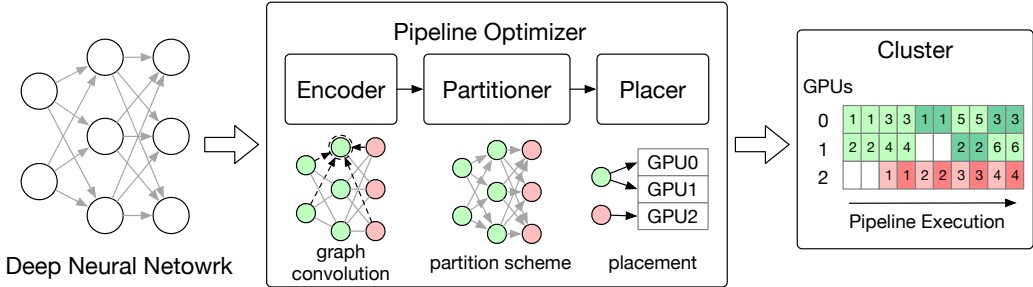

Figure 2: An overview of the architecture of *DRL-PP*.

keep track of outdated weights for back propagation, and gradient staleness is also introduced into DNN training.

In Figure 1b, all partitions in the pipeline have the same computation time for forward and backward, respectively. This is an ideal case in which the utilization of the accelerator can be maximized. However, it is difficult to partition a DNN model evenly in practice, especially when there are large layers in DNN models. To address this problem, PipeDream proposes to accelerate large partitions/layers through data parallelism. This allows a more flexible pipeline schedule and model partitioning scheme. Therefore, DNN models can be trained on multiple accelerators more efficiently.

## 2.2 CHALLENGES AND OPPORTUNITIES

Both GPipe and PipeDream assume DNN models are chain-structured. However, this is not true for many models, such as AlexNet, ResNeXt, and NASnet. Although PipeDream proposed that they could convert a multi-branching neural network into a chain structure by branch compression, this may result in some "large layers" in DNN models. Therefore, it is more challenging to load-balance these "large layers" in the pipeline.

Another problem is that PipeDream schedules the pipeline based on the profiled computation time of each layer in DNN models. PipeDream partitions a DNN model by measuring the computation time of each layer on a single accelerator. It computes the accumulated computation time of each partition by adding up the computation time of all layers in the partition. And then it estimates communication time by dividing activation size by bandwidth. Finally, PipeDream finds the optimal pipeline by solving a cost model-based partitioning problem with dynamic programming. However, profiling distributed machine learning training systems and workloads is difficult in practice. Even small errors can make a huge difference in pipeline placement. Thus, there is a need to design an efficient pipeline parallelism framework that optimize the pipeline schedule for training DNN models for different hardware environments agnostically.

## 3 PROPOSED FRAMEWORK: DRL-PP

To address the problems in the state-of-the-art pipeline parallelism frameworks, we propose a deep reinforcement learning (DRL) based pipeline parallelism framework, *DRL-PP*, that learns to optimize the pipeline partition for training large models across multiple accelerators distributedly. As Figure 2 illustrates, the core of *DRL-PP* is a DRL agent consisting of three components: encoder, partitioner, and scheduler, where the encoder describes the semantics of an operator in the computational graph, the partitioner partitions DNN models by walking through the computational graph node by node recurrently, and the scheduler assigns the partitions to accelerators smartly. By interacting with a real-world environment, *DRL-PP* gradually learns to partition and pipeline DNN models optimally.

### 3.1 GRAPH ENCODER

The DNN model can be represented by a computational graph where the nodes are operators, and the edge represents data flow between them. Many existing works (Lan et al., 2021b; Zhou et al., 2019; Addanki et al., 2018) propose using graph neural networks (GNNs) to learn comprehensive

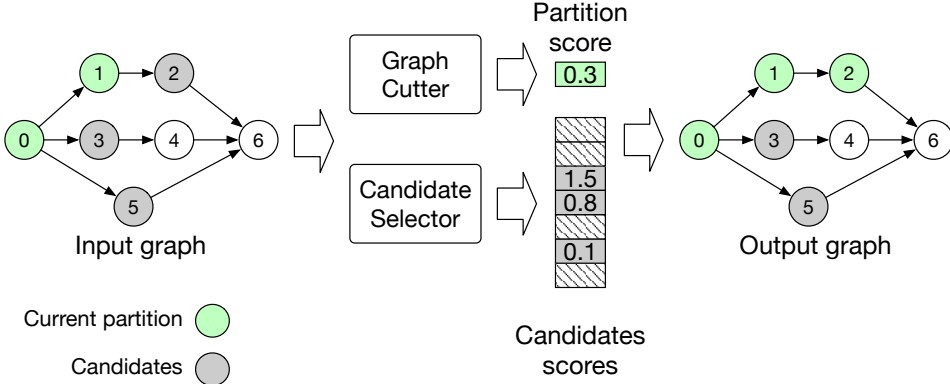

Figure 3: An illustration of the graph partitioner generating partitions recurrently.

representations of operators in a DNN model. Our framework also adopts this design in our DRL agent. We use a three-layer graph convolutional network (GCN) as the graph encoder. The formulation of each graph convolutional layer is described as follows:

$$\mathbf{GConv}(X, A) = \sigma \left( X\Theta || \tilde{D}^{-1} \tilde{A} X \Theta' \right) \tag{1}$$

where $\tilde{D}^{-1}$ denotes the degree matrix, $\tilde{A}$ is the adjacency matrix with inserted self-loops, $X$ represents the node feature matrix, $\sigma$ is a nonlinear function, $||$ is the concatenation operator, and $\Theta$, $\Theta'$ are the trainable parameters.

The node features fed into the graph encoder are the operator's type, input degree, input size, output degree, output size, and parameter size. The operator type is one-hot encoded, and the other features are normalized to range from 0 to 1. The adjacency matrix is a symmetric matrix, which means we add edges in both directions to make the graph undirected.

## 3.2 RECURRENT MODEL PARTITION ALGORITHM

Different from the existing pipeline parallelism frameworks which convert the DNNs into a chain before partitioning, *DRL-PP* partitions the DNNs directly over the computational graph. Although graph partitioning is a well-studied problem in the research community, it turns out that heuristics failed to find a satisfactory partition scheme for DNN models (Mirhoseini et al., 2017). And a fixed partition scheme also limits the flexibility of pipeline scheduling (Mirhoseini et al., 2018; Lan et al., 2021b). Inspired by XGNN (Yuan et al., 2020), we designed a recurrent model partitioner that generates partitions by walking through the computational graph node by node.

As Figure 3 illustrated, the partitioner consists of two components, a graph cutter and a candidate select The graph cutter computes a partition score based on the summary representation of current and previous partitions. The candidate selector computes the scores for all nodes that can be added to the current partition. In each recurrent step, the partitioner samples an action based on the calculated scores. This will be either adding a candidate node to the current partition or cutting the graph at the current node. The output graph will become the input graph in the next recurrent step.

The detailed recurrent graph partition algorithm of *DRL-PP* can be described by pseudocode as Algorithm 1, where LSTM($H, Seq$) is the LSTM layers that are initialized with hidden representation $H$ and the input data is sequence $Seq$, MLP is multi-layer perceptions that compute the partition scores and candidate scores.

As described in Algorithm 1, the partitioner first summarizes the partition $H_{G_k^s}$ by sum the representations of nodes $H_{v_i}$ in each partition. Then, the graph cutter encodes the summary of the current partition into a hidden representation $\tilde{H}_{G_k^s}$ by an LSTM layer. It computes the partition score $S_{G_k^s}$ by an multi-layer perception (MLP). The partition score indicates how good the current partition is; the higher the score, the more likely it is for the partitioner to take cut action. Next, the candidate selector encodes the representation of nodes in the current partition $\{H_{v_j}, \forall v_j \in G_k^s\}$ and the candidate node

---

**Algorithm 1** Recurrent Graph Partition Algorithm

---

**Require:** $H$ — node representations, $G$ — computational graph, $K$ — number of partition, $V_c$ initial candidate nodes, $T$ temperature of encouraging exploration

**Ensure:** Subgraphs $\{G_1^s, G_2^s, \cdots, G_K^s\}$.

1: **Initialize:** representation of empty partition $\tilde{H}_{G_0^s} = [0, \cdots, 0]$, score of empty partition $S_{G_0^s} = 0$.
2: **for** $k = 1$ **to** $K - 1$ **do**
3:      Initialize new partition $G_k^s = \{v_s\}$
4:      **repeat**
5:          Add all predecessors into the current partition $G_k^s = G_k^s \cup predecessors(G_k^s)$
6:          Add all successors into candidate nodes $V_c = V_c \cup successors(G_k^s) \setminus \{G_1^s, \ldots, G_k^s\}$
7:          **if** $V_c$ is $\emptyset$ **then**
8:              Cut the graph by set $S_{a_{cut}} = \infty$
9:          **else**
10:            Compute score of cut action $S_{a_{cut}} = S_{G_{k-1}^s} - T$
11:          **end if**
12:          Compute summary of $k$-th partition $H_{G_k^s} = \sum_{v_i \in G_k^s} H_{v_i}$
13:          Compute encoded representation of $k$-th partition $\tilde{H}_{G_k^s} = \text{LSTM}(\tilde{H}_{G_{k-1}^s}, \{H_{G_k^s}\})$
14:          Compute score of $k$-th partition $S_{G_k^s} = \text{MLP}(\tilde{H}_{G_k^s})$
15:          Compute score of candidates $S_{v_i} = \text{MLP}(\text{LSTM}(\tilde{H}_{G_{k-1}^s}, \{H_{v_j} : \forall v_j \in G_k^s\} \cup \{H_{v_i}\}))$
16:          Compute policy of actions $\mathbb{P} = Softmax(\{S_{a_{cut}}\} \cup \{S_{v_i} : \forall v_i \in V_c\})$
17:          Sample an action $a$ from the policy $\mathbb{P}$
18:          Update candidate nodes $V_c = V_c \setminus \{a\}$
19:          Update current partition $G_k^s = G_k^s \cup \{a\}$
20:      **until** $a == a_{cut}$
21:      Cut graph and sample a new start node from policy $v_s = Sample(\mathbb{P})$
22:      Update candidate actions $V_c = V_c \setminus \{v_s\}$
23: **end for**
24: Let the left of nodes become last partition $G_K^s = G \setminus \{G_1^s, \ldots, G_{K-1}^s\}$
25: **Return** subgraphs $\{G_1^s, G_2^s, \cdots, G_K^s\}$.

---

$\{H_{v_i}\}$ into hidden representation $\tilde{H}_{v_i}$, and feeds it into the MLP to compute the candidate scores $S_{v_i}$.

Finally, we apply the softmax function to the partition score $S_{G_k^s}$ and candidate scores $S_{v_i}$ to get the action policy P. We sample an action $a$ for each recurrent step, and then update the current partition $G_k^s$ and the candidate nodes $V_c$ accordingly. The partitioner performs the recurrent step repeatedly until all nodes are partitioned.

Note that we limit the max number of partitions $K$ equal to the number of accelerators. So we stop partitioning at the $(K - 1)$-th cut, and the rest of the nodes will be the last partition. However, the partitioner may generate fewer partitions than the number of accelerators. In this case, some partitions will be assigned to multiple accelerators by the pipeline scheduler. Similar to data parallelism, the parameters of these partitions will be replicated on multiple accelerators.

### 3.3 PIPELINE SCHEDULER

With the partition scheme generated by the model partitioner, we now need to assign partitions to accelerators. We first generate a learnable embedding for each accelerator, called accelerator embedding. Then, we sum the node representations in each partition to get summary representation of partitions. As Figure 4a shows, we concatenate the embeddings of accelerators and summary representations of partitions in pairs and feed them into a MLP, which computes a score for each accelerator-partition pair. As Figure 4b shown, we apply the softmax function to the scores of partitions accelerator by accelerator (column-wise). By doing so, each accelerator only holds one partition. It is possible that some partitions are not assigned to accelerators. In this case, we will penalize the pipeline scheduler with a negative reward and sample a new placement. Figure 4c and 4d show how the model partitions are placed on the accelerators and executed with the pipeline schedule

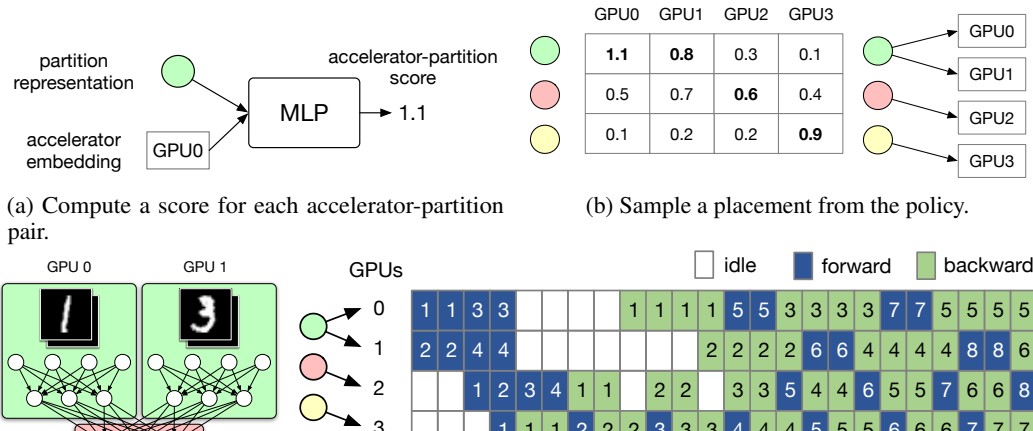

(a) Compute a score for each accelerator-partition pair.

(b) Sample a placement from the policy.

(c) Place model partitions on accelerators.

(d) Pipeline schedule of generated placement and partition scheme.

Figure 4: An illustration of the pipeline scheduler generating placement with given model partitions and accelerators.

with the generated placement and partition scheme. In this example, the green partition is replicated on GPU 0 and 1, and trained with data parallelism.

### 3.4 TRAINING WITH DEEP REINFORCEMENT LEARNING

In *DRL-PP*, we train all three components jointly with the proximal policy optimization (PPO) (Schulman et al., 2017) algorithm to gradually learn a better policy network. We sample multiple placements for each partition scheme generated by the model partitioner. We evaluate all valid placements (each partition are assigned to at least one accelerator) in a real-world environment and measure the per mini-batch runtime for each placement. We use the negative natural logarithm of the per mini-batch runtime as the reward:

$$R_{i,j} = -\ln r_{i,j} \tag{2}$$

where $r_{i,j}$ is the per mini-batch runtime of training DNN models with placement $j$ and partition scheme $i$. For the reward of the partition scheme, we use the average of the rewards of placements sampled from the same partition scheme.

## 4 EVALUATION

### 4.1 BENCHMARKS AND BASELINES

We selected five typical DNN models from image classification and natural language processing tasks as our benchmarks for evaluation:

**Image Classification Task**. We chose four popular image classification models as benchmarks, ResNet-50 (He et al., 2016), VGG-16 (Simonyan & Zisserman, 2015), AlexNet (Krizhevsky et al., 2012), and ResNeXt-101 (Xie et al., 2017). These image classification models are also used as benchmarks in PipeDream's experiments. We use the same per-GPU mini-batch size of 64 for all image classification models.

**Natural Language Processing**. We use the 4 LSTM layers version of Google's Neural Machine Translation (GNMT) with an attention layer as a benchmark, where each LSTM layer has 256 hidden units. The sequence length is limited to between 20 and 50 words. The per-GPU mini-batch size is set to 128, where the model can fit into a single GPU.

We compare the performance of *DRL-PP* to three baselines:

Table 1: Per mini-batch runtime (in seconds) of placements found by different approaches.

| MODELS | DP | GPIPE | PIPEDREAM | DRL-PP | SPEEDUP OVER | |
|---|---|---|---|---|---|---|
| | | | | | DP | PIPEDREAM |
| RESNET-50 [14] | **0.051** | 0.158 | **0.051** | **0.051** | 1× | 1× |
| VGG-16 [25] | 0.165 | 0.193 | 0.107 | **0.082** | 2.01× | 1.3× |
| ALEXNET [16] | 0.057 | 0.068 | 0.012 | **0.010** | 5.7× | 1.2× |
| RESNEXT-101 [27] | 0.125 | 0.297 | 0.125 | **0.116** | 1.08× | 1.08× |
| GNMT-4 [26] | 1.082 | 1.141 | 0.841 | **0.689** | 1.57× | 1.22× |

**Data Parallelism (DP) and GPipe.** We use the data parallelism module (pyt, 2022c) and GPipe (Huang et al., 2019) provided by PyTorch to train the benchmarking models across multiple accelerators. We use the NCCL backend to achieve the best performance (NCC, 2022; pyt, 2022b). Different from *DRL-PP* and PipeDream, both data parallelism and GPipe use synchronized weight updates, which means there is no staleness introduced during training.

**PipeDream.** We use the open source code published by the authors on the GitHub (pip, 2019) to reproduce the results on our experimental platform. And for multi-branching deep neural networks, we compress the branches using the script provided in their git repo. Note that PipeDream was implemented with the Gloo backend (Glo, 2022), which is known to be slower than the NCCL backend. The reason they didn't use the NCCL backend is that pipeline parallelism uses point-to-point communication operations to send and receive model activations between accelerators. And the NCCL backend requires explicit synchronization to perform point-to-point communication operations. PipeDream does not synchronize communications between accelerators. Hence, we use the Gloo backend for PipeDream in the experiments.

## 4.2 EXPERIMENTAL SETUP

We implement our framework *DRL-PP* with PyTorch (pyt, 2022a) and evaluate all baselines using the following settings[1]:

**Architecture of the DRL Agent.** In our framework, the graph encoder consists of three-layers of GCNs with 32, 64, and 128 hidden units respectively, the graph partitioner consists of two one-layer perceptions with 128 hidden units and two LSTM layers with 128 hidden units, and the pipeline scheduler is two-layer MLP with 128 hidden units. The size of accelerator embeddings is 128.

**Staleness and Weight Update.** To be a fair comparison, we implemented the same update mechanisms as PipeDream (as shown in Figure 1b and 4d ), which maintains all versions of activations for different mini-batches. As a result, the staleness introduced by *DRL-PP* is the same as PipeDream. Therefore, the performance of benchmarking models trained with *DRL-PP* and PipeDream is similar. The only difference is the per mini-batch runtime caused by the different pipelining schedules.

**Performance Evaluation Metric.** We evaluate placement performance by measuring the per mini-batch runtime of training the benchmarking models. To be accurate as possible, the per mini-batch runtime is averaged over 30 mini-batches during the DRL agent training, and 1000 mini-batches in the final evaluation.

## 4.3 RESULTS AND ANALYSIS

We summarize the results of a comparison of *DRL-PP* with data parallelism, GPipe and PipeDream for all benchmarks in Table 1, and analyze the results as follows"

**GPipe.** In all benchmarks, GPipe is the slowest framework among all baselines. This is because GPipe is designed based on model parallelism. It is typically used for training large models that cannot fit into a single GPU, where data parallelism cannot be implemented. It uses synchronized weight

---

[1]Hardware & software environment for evaluation and other hyperparameters for DRL training are listed in the Appendix.

update mechanisms to avoid introducing any staleness during the training. Thus, the accelerators are not fully utilized in the pipeline schedule.

**PipeDream and DRL-PP.** As Table 1 shows, our framework *DRL-PP* either outperforms or achieves the same performance as data parallelism and other pipeline parallelism frameworks in all benchmarks. For ResNet-50, both *DRL-PP* and PipeDream achieved the same performance as data parallelism. The reason is that ResNet-50's activation size is too large, and that the communication overhead is more significant than the benefits of pipeline parallelism. As a result, both methods failed to find a better pipeline scheme than data parallelism for ResNet-50. For other benchmarks, they all find a better placement than data parallelism with significant speedup.

For VGG-16, AlexNet and GNMT-4, both PipeDream and *DRL-PP* find similar placements. They partition VGG-16 and AlexNet into two stages and replicate the first large stage on 3 GPUs with data parallelism. They place the second stage on the last GPU. For GNMT-4, they both partition it into four stages and place each stage on a GPU. *DRL-PP*, however, finds a better pipelining placement than PipeDream for GNMT-4. This is because GNMT-4 is a multi-branch DNN. PipeDream compresses the DNN branches into a "large layer". Hence, the computational graph is converted into a chain structure and placed as other chain-structured DNNs. In contrast, *DRL-PP* views multi-branched DNNs as a graph and partitions the model inside the branches. As the experimental results show, *DRL-PP*'s pipeline placements are $1.22\times$ faster than PipeDream's for GNMT-4.

**Placement Analysis.** We analyze the placement found by DRL-PP and PipeDream. For ResNet-50, both methods find that data parallelism is the best placement and model parallelism does not speed up the training. Both methods divided VGG16 into two stages, where the first stage is replicated on 3 GPUs, and their outputs are aggregated on the last stage. This placement is referred as "3-1" placement. The only difference between them is that *DRL-PP* partitioned 2 more layers into the first stage than PipeDream, which makes the partitions more balanced. We also observed similar trends on AlexNet and ResNeXt-101. It is due to the fact that the last half of the model has many more parameters, while the computation time is much shorter than the first half, that these models have been partitioned into "3-1" placements. Thus, stage replication is applied to the first half of the model to speed up the computation. The parameters of the last half of the model are stored on a single GPU, to avoid the overhead of parameter synchronization between GPUs.

We notice that PipeDream's speedup over data parallelism is less significant than reported in PipeDream (Narayanan et al., 2019), especially for VGG-16. There are two reasons. First, our experiments use fewer accelerators. PipeDream used 16 GPUs while we only used 4 GPUs. Second, the placement of VGG-16 is "15-1", which means the size of partitions is highly unbalanced. In our experiments, we found that the placement generated by *DRL-PP* for VGG-16 was also highly unbalanced. As we only have 4 GPUs, the 3-1 placement is the most optimal placement on our cluster.

For the GNMT-4 model, both DRL-PP and PipeDream find that "1-1-1-1" placement (model parallelism) is the most effective placement. The reason is that, in GNMT-4 model, the activations size between layers is much smaller than the parameters size. Thus, the communication cost of model parallelism is less than data parallelism and there is no room to speed up the training by stage replication.

Another observation is that the placements generated by *DRL-PP* and PipeDream are very similar. PipeDream, however, requires accurate profiling of DNN training workloads and system performance, including the computation power of accelerators and bandwidth between them. This may not be feasible in practice. And even a small profiling inaccuracy can result in a significant placement drop.

**Convergence of DRL Agent.** We investigate the training process of *DRL-PP* in optimizing the pipeline schedule for the VGG-16 model. Figure 5 illustrates the training curve of *DRL-PP*'s DRL agent, which indicates that it can efficiently discover near-optimum placements within 200 training steps. Subsequently, *DRL-PP* gradually explores and improves the quality of placements, finally converging at the 600th step. The entire training process takes approximately 4 hours, which is negligible compared to the actual training time required for VGG-16 over ImageNet.

**Staleness and Convergence of DNN Training.** PipeDream and DRL-PP use asynchronous weight updates to accelerate pipeline execution, which also introduces staleness in DNN training. To analyze the influence of staleness on model convergence, we measured the wall-clock training time and

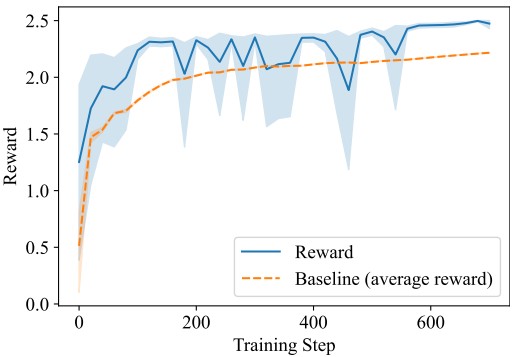
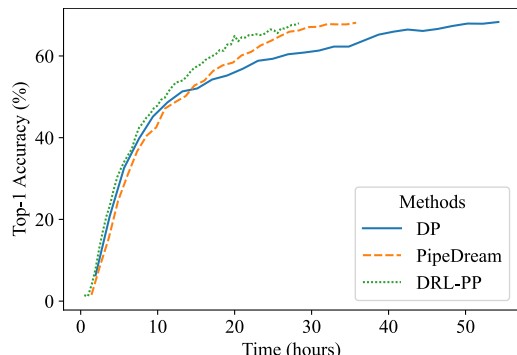

Figure 5: The reward and baseline (average reward) of training DRL-PP agent for optimizing the pipeline schedule of VGG-16.

Figure 6: Top-1 accuracy vs. time for VGG-16 training on ImageNet dataset with different methods.

number of epochs of different methods to train VGG16 model to target validation top-1 accuracy of 68% on ImageNet dataset. As Figure 6 shows, *DRL-PP* takes about 28.4 hours for VGG16 to converge to the target validation accuracy, which is 1.26× and 1.91× faster than PipeDream and data parallelism respectively. The speedup ratio is slightly less than the results in Table 1, the reason is that *DRL-PP* takes a few more epochs to train VGG16 to the target accuracy. Specifically, *DRL-PP* takes 60 epochs to reach the target validation accuracy, which is 2 epochs more than PipeDream and 4 epochs more than data parallelism.

## 5 MORE RELATED WORKS

**Pipeline Parallelism.** PipeDream-2BW (Narayanan et al., 2021) is a variant version of PipeDream that focusing on memory efficiency of the pipeline parallelism. Its double-buffered weight update (2BW) and flush mechanisms ensure high throughput, low memory footprint, and weight update semantics similar to data parallelism. They has shown great efficiency on optimizing transformer-based language models. i.g., BERT and GPT.

HetPipe (Park et al., 2020) improves the pipeline parallelism by considering the heterogeneity of devices when partitioning the workloads. It groups a mixture of devices into a virtual worker such that each worker has similar computational resources, and then partition and pipeline the neural network across multiple virtual workers.

**Device Placement.** Mirhoseini *et al.* (Mirhoseini et al., 2017) proposed to use of a DRL agent to generate the model parallelism scheduling plan (device placement) for deep neural networks. Hierarchical Planner (Mirhoseini et al., 2018), Spotlight (Gao et al., 2018a), Placeto (Addanki et al., 2018), Post (Gao et al., 2018b), and EAGLE (Lan et al., 2021a) followed this idea and proposed many advanced DRL agent architecture to improve the quality of generated device placement. GDP (Zhou et al., 2019) and Mars (Lan et al., 2021b) further improves the generalizability of the agent. Thereby, they do not need to re-train the DRL agent from scratch for the unseen machine learning workloads.

## 6 CONCLUDING REMARKS

In this paper, we introduce a DRL-based pipeline parallelism framework *DRL-PP*. The core of *DRL-PP* is a DRL agent consisting of a graph encoder, a recurrent model partitioner and a pipeline scheduler. *DRL-PP* has several advantages: it is agnostic to the machine learning cluster architecture (except the number of accelerators), partitions the DNNs with a graph view, does not require branch compression for partitioning the multi-branching DNN models, compatible with asynchronous pipeline weight updates proposed by PipeDream, enjoys the benefit of the most efficient distributed communication backend NCCL. From the experimental results, *DRL-PP* can speed up benchmark models' training by up to 6.8× faster over data parallelism and 1.3× faster than PipeDream.

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
