## A  HARDWARE ENVIRONMENT

We optimize the benchmarking models on a physical machine, which has 4 NVIDIA P100 GPUs (each has 16GB RAM), 2 Intel E5-2630 CPUs, and 128GB memory. We didn't test our framework with cross-machine scenario, but our framework do not require any knowledge of the environment except the number of accelerators.

## B  SOFTWARE ENVIRONMENT

We use the PyTorch version 1.12.1, with Python 3.9.6, CUDA 11.4.2 and NCCL 2.11.4 and CUDNN 8.2.0. We made an minor modification of PyTorch to disable the gradients version check feature introduced since PyTorch 1.5.0. The modification is very simple, we only need to change the condition of a `if` statement in `saved_variable.cpp` (sav, 2022) always be `True`, thus PyTorch will skip the gradient version check. This modification is also required by PipeDream if they are using the latest version of PyTorch.

## C  HYPERPARAMETERS OF DRL TRAINING

We train *DRL-PP* with deep reinforcement learning algorithm proximal policy optimization (PPO). The objective function of PPO is shown in Eq. (6):

$$A_{i,j} = \overline{R} - R_{i,j} \tag{3}$$

$$r\left(\theta\right) = \frac{\pi_\theta}{\pi_{\theta_{\mathrm{old}}}} \tag{4}$$

$$L^{\mathrm{TRPO}}\left(\theta\right) = \mathbb{E}\left[r\left(\theta\right)\hat{A}_{i,j}\right] \tag{5}$$

$$L^{\mathrm{CLIP}}(\theta) = \mathbb{E}[\min(r(\theta)\hat{A}_{i,j}, \mathrm{clip}(r(\theta), 1-\epsilon, 1+\epsilon)\hat{A}_{i,j})] \tag{6}$$

where $\hat{A}_{i,j}$ is estimated advantage of placement $(i,j)$ by subtracting the average of history reward $\overline{R}$ from its reward $R_{i,j}$. $\pi_\theta$ is the policy based on current parameters $\theta$, $\theta_{\mathrm{old}}$ is the parameters before updating. To avoid an excessively large policy update, PPO uses clipped surrogate objective over trust region policy optimization (TRPO) in Eq. 5. Here, $\epsilon$ is a hyperparameter for adjusting the clip region.

During the DRL agent training with PPO algorithm, we sample 10 placements from each partition scheme generated by the graph partitioner. For every 20 placements, we shuffled them into four mini-batches and performed updates on each of the individual mini-batches. After repeating this for three epochs, the DRL agent will generate new policies with updated parameters. For other hyperparameters of PPO algorithm, we set the clip ratio $\epsilon$ of 0.2, and the coefficient of entropy is set to 0.001. We use Adam optimizer with a learning rate of 0.0003 and gradient clipping with a 1.0 norm.

## D  MORE EXPERIMENTAL RESULTS

As Table 2 shows, it takes 60 epochs for *DRL-PP* to train VGG16 to the target validation accuracy, which is 2 epochs more than PipeDream and 4 epochs more than data parallelism. However, due to the faster per-mini-batch runtime of *DRL-PP* compared to data parallelism, the wall clock training time of *DRL-PP* is almost half that of data parallelism.

Table 2: Training time (in hours) and epochs of VGG16 on ImageNet to 68% Top-1 accuracy.

| METHODS | DP | PIPEDREAM | DRL-PP |
|---|---|---|---|
| # OF HOURS | 54.26 | 35.79 | **28.35** |
| # OF EPOCHS | **56** | 58 | 60 |

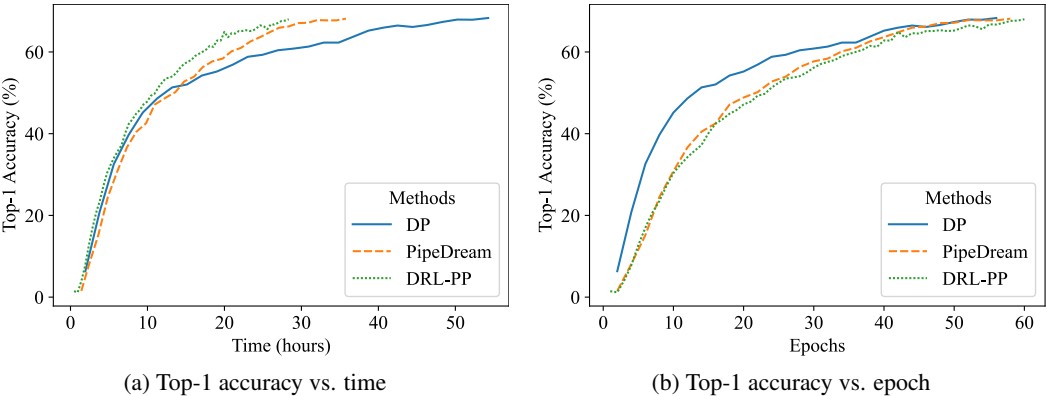

(a) Top-1 accuracy vs. time

(b) Top-1 accuracy vs. epoch

Figure 7: Training curve of VGG-16 on ImageNet dataset with different methods.

As the training curve shown in Figure 7b, we observed that *DRL-PP* and PipeDream have similar accuracy vs. epochs curve, which means their converge speed is almost the same. In the first 40 epochs, data parallelism converges faster than both DRL-PP and PipeDream significantly. This is because data parallelism does not introduce any staleness in the training, thus the quality of the model weights update is better. In the last 20 epochs, the lead that data parallelism had gained earlier gradually became negligible, and eventually all methods achieved the same target validation accuracy, 68% Top-1 accuracy.

## E    FUTURE WORKS

There are a few possible improvements that can be implemented over *DRL-PP* in the future:

**Heterogeneity.**  In the pipeline scheduler design, we assigned a learnable embedding for each accelerator. We learn the heterogeneity of the cluster in these embeddings. For example, we can represent the cluster structure as a graph, where the node features are the information of accelerators and the edges are inter-connections. Learning with a graph neural network, we can easily obtain a comprehensive embedding encoded with the heterogeneity of the accelerators, including computation power, the bandwidth of inter-connection, and memory constrain. Thus, the DRL agent will be able to place the partitions with the knowledge of the heterogeneity of the cluster like HetPipe (Park et al., 2020).

**Pre-train and Generalization.**  Suggested by Placeto (Addanki et al., 2018), GDP (Zhou et al., 2019), and Mars (Lan et al., 2021b), it's possible to train a powerful DRL agent that can generate high-quality placements for unseen DNNs with few-shot training. As PipeDream used a simulator to estimate the placement quality in their optimizer, it's also an excellent idea to pre-train the DRL agent with a simulator and fine-tune it with real-world samples. With the pre-training and the generalization, we can significantly reduce the training overhead of the DRL agent.

**Memory Efficiency.**  PipeDream2BW (Narayanan et al., 2021) reduce the memory footprint by using double-buffered weight update and flush mechanisms. This mechanism also can be applied to *DRL-PP*, since we are using a similar weight updates scheme for pipeline parallelism.

## F    LIMITATION

As mentioned in Section B, both our framework *DRL-PP* and PipeDream require disable gradient version check by modifying PyTorch source code. This may limit the usage of our framework, but it is possible to be avoid by adopting double buffered weight update scheme proposed by PipeDream-2BW (Narayanan et al., 2021). It also can be solved by re-implement *DRL-PP* with more advanced distributed training frameworks, such as DeepSpeed.