# OpenReview forum: "Pipeline Parallelism Optimization with Deep Reinforcement Learning"
_ICLR.cc/2024/Conference — Submitted to ICLR 2024_

### Official Review · Reviewer_VkLQ · 2023-10-26

**Soundness:** 3 good
**Presentation:** 3 good
**Contribution:** 2 fair
**Rating:** 5
**Confidence:** 4

**Summary:**

This paper presents a method to optimize the widely used pipeline parallelism using deep reinforcement learning. Different from previous work on pipeline parallelism, the presented method is capable of generating partition for models which are not fully chain structured without introducing branching compression. Moreover, the method does not require a profiling procedure which could introduce significant error in the real deployment of training.  A graph encoder, a recurrent model partition algorithm and a pipeline scheduler are introduced to facilitate the method. Evaluation on the policy training and comparison with data parallelism/ other pipeline parallelism methods demonstrate the effect of the paper.

**Strengths:**

This paper presents a method to optimize the widely used pipeline parallelism using deep reinforcement learning. Different from previous work on pipeline parallelism, the presented method is capable of generating partition for models which are not fully chain structured without introducing branching compression. Moreover, the method does not require a profiling procedure which could introduce significant error in the real deployment of training.  A graph encoder, a recurrent model partition algorithm and a pipeline scheduler are introduced to facilitate the method. Evaluation on the policy training and comparison with data parallelism/ other pipeline parallelism methods demonstrate the effect of the paper.

**Weaknesses:**

The main contribution of the paper is the introduction of deep reinforcement learning to optimize the challenge on pipeline parallelism. However, it brings extra overhead on determining the pipelining strategy. Moreover the evaluation of the method is not enough, especially when considering pipeline parallelism, we expect the author could demonstrate the effect of the proposed method on large-scale systems. The models and hardware configurations of the evaluation is far below satisfactory. In particular, the improvement compared to PipeDream is not based on the settings of the PipeDream paper. Last but not least, the authors does not compare to some more recent proposed approaches on pipeline parallelism.

**Questions:**

1. I wonder the sclalebility of the proposed approach, when model and computing resource scales increase.
2. Different levels of communication demonstrate huge margins on the communication capability. For instance, NVlink/ PCIe connections between GPU cards, or Ethernet/ RDMA connections between machines. The pipeline/data/tensor parallelism design should be compatible with the network/ model configuration. Is it possible to generalize the proposed approach  to  the hybrid parallelism optimization on heterogeneous network configurations?

---

### Official Review · Reviewer_nNSt · 2023-10-30

**Soundness:** 2 fair
**Presentation:** 2 fair
**Contribution:** 3 good
**Rating:** 5
**Confidence:** 3

**Summary:**

Summary:
-----------------
This paper proposes a deep reinforcement learning-based pipeline parallelism framework called DRL-PP for optimizing distributed training of large-scale deep neural networks. The core of DRL-PP is a DRL agent consisting of three components: a graph encoder, a recurrent model partitioner, and a pipeline scheduler. The graph encoder uses GCN to encode semantic information of operators in the computational graph. The recurrent partitioner generates model partitioning by traversing each node of the graph recursively. The pipeline scheduler assigns partitions to GPU devices with optimized pipeline placement. DRL-PP is trained end-to-end using PPO to learn the optimal policy. Experiments show that DRL-PP can accelerate distributed training on various benchmarks compared to data parallelism and PipeDream.

**Strengths:**

Strengths:
-----------------
+	Handles complex model structures without simplifying to chains, more flexible partitioning.
+	Integrates well with asynchronous pipeline execution in PipeDream.
+	End-to-end learning allows jointly optimizing all components for pipeline parallelism.

**Weaknesses:**

Weaknesses:
-----------------
-	Performance evaluation is weak (see detailed in C1).
-	Require a large number of trails to converge to optimal policy (see detailed in C2).
-	Performance gains highly depend on model structures, less effective for some networks  (see detailed in C3).

**Questions:**

Comments:
-----------------
C1:  Firstly, the experiments only evaluate DRL-PP on five benchmark models - four image classification models and one machine translation model. It remains unclear how well DRL-PP would perform for larger and more complex models. The authors should investigate the computational overhead of using DRL-PP to search pipeline parallelism placements for larger models, such as BERT and GPT-3. There may be scalability issues that are not uncovered by the small benchmark models. Secondly, the experimental evaluation is too narrow, evaluating only on the VGG-16 model. To demonstrate the effectiveness of the proposed DRL-PP framework, the authors need to conduct experiments on a broader set of models. In particular, the paper only shows convergence analysis and accuracy results for VGG-16 on ImageNet. The authors claim that DRL-PP can accelerate the training of deep neural networks, but this is only convincingly shown for one model. To substantiate this claim, the authors should evaluate the training accuracy and convergence when using DRL-PP optimized pipeline parallelism for multiple models, including CNNs, transformers, RNNs.

C2:	My main question about this work is - does DRL-PP need to search for the pipeline parallelism policy from scratch every time a new model is introduced? The paper mentions that DRL-PP is trained using deep reinforcement learning to optimize the pipeline placement and partitioning for a given model. However, it is unclear if this training has to be repeated for each new model, or if the policies learned by DRL-PP can generalize across models. If DRL-PP has to re-learn the pipeline parallelism policy for every new model, it would be concerning in terms of computational overhead. For instance, if I want to train a BERT model, does that mean DRL-PP has to go through many iterations of searching the policy space before reaching an optimal pipeline strategy? This trial-and-error search may become prohibitively expensive for those much larger models. The authors should discuss how the policies learned by DRL-PP could potentially transfer or generalize to unseen models. Is the graph encoder representing models in a way that allows generalization? Can policies for similar models be reused? Or is DRL-PP only practically viable for fixed benchmark models that are used to pre-train the policies? Understanding the model-dependence and computational overhead of policy search would help evaluate the real-world usefulness of DRL-PP. More analysis is needed about the efficiency and generalization ability of the framework beyond small benchmark experiments.

C3:	The results in the paper show that the performance gains of DRL-PP vary significantly across different model architectures. For example, it has almost no speedup for ResNet-50, but 2x faster for VGG-16. This is because pipeline parallelism relies heavily on model structures. For some models, if they have very large intermediate activations, the communication cost could outweigh computation, making pipeline parallelism less efficient. While for models with more layers and parameters, pipeline parallelism can lead to noticeable speedup. Therefore, the advantages of pipeline parallelism depend on model structures. For certain models it is less effective, requiring more custom optimization. This is a limitation of DRL-PP.

C4:	While this paper proposes a novel DRL-based pipeline parallelism framework, it does not sufficiently differentiate itself from prior arts like PipeDream. The authors state that existing techniques rely on profiling DNN performance on the cluster to optimize pipeline parallelism. They posit that this may not be feasible in practice. However, the paper does not comprehensively analyze the limitations of prior arts or empirically demonstrate scenarios where profiling-based methods fail. Simply asserting that profiling has drawbacks is not enough to motivate DRL-PP.

---

### Official Review · Reviewer_3JAq · 2023-11-01

**Soundness:** 2 fair
**Presentation:** 2 fair
**Contribution:** 1 poor
**Rating:** 3
**Confidence:** 5

**Summary:**

This paper starts from the challenge of partitioning a deep learning model with millions of parameters in order to run it efficiently on various devices such as a cluster of accelerators, e.g., GPUs and TPUs. The main idea is to distribute the training of deep neural network (DNN) models via the pipeline parallelism. For this purpose, this paper presents a deep reinforcement learning (DRL)-based pipeline parallelism (DRL-PP) framework, that learns to optimize the pipeline schedule for training large DNN models across multiple accelerators. The DRL-PP consists of a graph encoder, describing the semantics of an operator in the computational graph, followed by a recurrent model partitioner and a pipeline scheduler that learns to partition and place operations on various GPU devices automatically.

**Strengths:**

+ Pipeline parallelization is important for improving performance

**Weaknesses:**

- Lack of a formal analysis of the problem
- Incomplete experimental analysis
- Unclear statement and incomplete description of related work.

**Questions:**

While efficient processing of deep learning models is required, the following major issues need to be addressed:
1) There seems to be a discrepancy in the introductory text because the very first sentence talks about DNNs and the next sentence “most state-of-the-art image classification models and natural language processing models (Brown et al., 2020; Zhai et al., 2022) have billions of parameters and take days or even weeks to train to satisfactory accuracy” and those involve other ML techniques not just DNNs. This inconsistency is not major but it would be good to provide a coherent statement about the size of the DNNs or focus the paper on the general image and NLP approaches.
2) I am not sure if within the computer engineering and systems community GPUs are considered accelerators. Accelerator has a very specific definition. Again this may sound / be minor but an accelerator refers to a specific hardware for a specific task like speech processing, etc.
3) Data parallelism and computation parallelism have been studied for a long time way before the mentioned paper of Krizhevsky et al., 2012 , for example there are many sessions each year or edition of major conferences in computer architecture, embedded systems and design automation venues on just these two topics either alone or treated somehow holistically. It is unfair to cite a paper from 2012 when there are other around 2000… For example, how do we model data parallelism? Or computation parallelism? What models of computation exist and were used in 1990s and before?
4) In Section 3.3, "Pipeline Scheduler," additional information is required for clarity and comprehension. Firstly, the features of the accelerator embeddings need to be explicitly defined. The authors should provide details on the representation of various accelerators, such as GPUs and TPUs, using accelerator embeddings. Secondly, while the authors discuss allocating different model partitions to various accelerators, they do not explain how a specific pipeline schedule is subsequently generated. To address this gap, the authors should elucidate the process involved in determining and creating the pipeline schedule.
5) The authors state that they used proximal policy optimization (PPO) algorithm to train all three components in DRL-PP jointly to gradually learn a better policy network. Among three components, there are many neural network layers, such as GCN layers in graph encoder, LSTM layers in recurrent partitioner and MLP layers in pipeline scheduler. However, while the authors presented the final reward function, they did not elaborate on the training details of these neural network layers. The authors should provide the loss functions for the neural networks and demonstrate how the weights of the neural network layers are updated based on the final reward.
6) In the experiments section, details regarding the devices used for execution should be provided. Additionally, as the authors discuss in Section 5, "More Related Works," PipeDream-2BW and HetPipe are two recent significant advancements in the field. The authors should justify their decision to exclude these two works from their experimental comparisons, ensuring a comprehensive evaluation and context for the presented results.
7) Distributed strategies and pipeline parallelism has been considered, developed and evaluated well before Huang et al (neurIPS 2019) for a wide range of computational benchmarks that some happen to include machine learning models while these benchmarks also involve high performance computing examples like molecular dynamics simulations , etc… that typically run on servers and supercomputing but with the advent of hundreds of heterogeneous processing elements on a single chip have also been considered on mobile systems or edge computing devices. Here is an example of one of the very first pipeline parallelization "A load balancing inspired optimization framework for exascale multicore systems: A complex networks approach." In 2017 IEEE/ACM International Conference on Computer-Aided Design (ICCAD), pp. 217-224. IEEE, 2017 that provides a comprehensive analysis of both data and computation parallelism actually extracting it from the software through advanced static and dynamic compiler techniques and some other mathematics that I am not fully aware. Here are a few more related papers Google scholar reveal:  "Exploiting coarse-grained task, data, and pipeline parallelism in stream programs." ACM SIGPLAN Notices 41, no. 11 (2006): 151-162. "Analytical modeling of pipeline parallelism." In 2009 18th international conference on parallel architectures and compilation techniques, pp. 281-290. IEEE, 2009. "On-the-fly pipeline parallelism." ACM Transactions on Parallel Computing (TOPC) 2, no. 3 (2015): 1-42. "Self-optimizing and self-programming computing systems: A combined compiler, complex networks, and machine learning approach." IEEE transactions on very large scale integration (VLSI) systems 27, no. 6 (2019): 1416-1427. "Plasticity-on-chip design: Exploiting self-similarity for data communications." IEEE Transactions on Computers 70, no. 6 (2021): 950-962. "A design methodology for energy-aware processing in unmanned aerial vehicles." ACM Transactions on Design Automation of Electronic Systems (TODAES) 27, no. 1 (2021): 1-20. Here is a paper that provides the first distributed parallelization: "A distributed graph-theoretic framework for automatic parallelization in multi-core systems." Proceedings of Machine Learning and Systems 3 (2021): 550-568. In general, prior work should be more thoroughly covered and credits should be given to first of their kind papers.
8) The authors state “While the idea of distributing tasks in a pipelined fashion is not novel, there are still many challenges when applying it to DNN training tasks, especially for complex DNN models” which shows that they agree with the above major shortcomings but fail to specify the challenges. Memory interdependence, limited memory and communication overhead are just a few not to mention data structure issue, etc. Please check the prior / existing literature and cite these challenges that have been analyzed in the computer architecture and computer systems community for 20-30 years now.  For example how is the DNN training different from some computer architecture or high performance computing benchmarks… many deal with large matrix to matrix multiplication but also with much more in terms of computation complexity. Again, this has been discussed recently as it can be seen from some of prior works I could find on Google scholar. Are there any papers on deep reinforcement learning for parallelization and pipeline parallelism? How do they differ?
9) What do the authors mean by operators in this sentence “DNN model can be represented by a computational graph where the nodes are operators”? There have been works on compiler analysis as mentioned above, but there have also been works on learning code representations like “Learning code representations using multifractal-based graph networks” 2021 IEEE International Conference on Big Data (Big Data), and many others. What does the operator and the graph convolutional networks bring new compared to these existing code representations?
10) The authors state “The adjacency matrix is a symmetric matrix, which means we add edges in both directions to make the graph undirected.” But making the graph undirected seems to be counterintuitive because data flow graphs clearly encode how data flows and it is unclear what this unidirectional-ity brings as a benefit if any.
11) What is the computational graph? Why and how is this computational graph different from other representations of programs / software? Why transforming a code like the DNN into a chain  graph and then do partitioning and not partition it first and then do pipelining?
12)  Maybe reading these sentences “Although graph partitioning is a well-studied problem in the research community, it turns out that heuristics failed to find a satisfactory partition scheme for DNN models (Mirhoseini et al., 2017). And a fixed partition scheme also limits the flexibility of pipeline scheduling (Mirhoseini et al., 2018; Lan et al., 2021b).” written by the authors in relation to existing work on automatic parallelization for complex programs via centralized and distributed graph partitioning would give a pause to think and realize that maybe the problem is well-studied and good approaches exist.
13) Can the authors show what is the computational graph and some of its nodes and edges and how it differs from existing models of computations used in other papers?
14) It is not entirely clear how the Algorithm 1 Recurrent Graph Partition Algorithm works. For example , does it require to know the entire graph? It was mentioned in the introduction that a distributed approach will be provided yet the algorithm takes the whole computational graph it seems.
15) Also it seems that the Algorithm 1 is related to simulated annealing. Please correct me if I am wrong. Irrespective of this, it requires that the algorithm takes as input the number of partitions so this raises a question, are there algorithms for partitioning that do not require this? There have been some progress in differential geometry and particularly applications to graph partitioning. How is Algorithm 1 related or different from existing graph partitioning algorithms?
16) How do we enforce that the subgraphs {Gs1,Gs2, · · · ,GsK} returned by Algorithm 1 balanced or this is not enforced? How do they contribute to higher efficiency because if we do not choose carefully K the number of partitions we may provide a way suboptimal solution maybe worse than not doing it at all.
17) Since the computational graph is undirected, how does the Algorithm 1 ensure the correctness of the computation? Is it the case that sometimes can create more communication than needed? How should we choose K to minimize the communication overhead?

**Details Of Ethics Concerns:**

Not applicable.

---

### Official Review · Reviewer_dWCJ · 2023-11-04

**Soundness:** 3 good
**Presentation:** 2 fair
**Contribution:** 2 fair
**Rating:** 5
**Confidence:** 3

**Summary:**

This paper presents a deep reinforcement learning (DRL)-based pipeline parallelism framework, DRL-PP.
The work comprises three components (1) graph encoder to parse the computational graph, (2) recurrent model partitioner to partition the graph, and (3) pipeline scheduler to place the operations on various GPU devices.

The paper builds on the concept of pipeline parallelism explored in GPipe and PipeDream to build a RL based method to improve the performance of DNN training.
The paper seems to make a set of reasonable decisions that leads to good performance.
Considering, better utilization of pipeline parallelism of DNN training can lead to large benefits such as less cost for training, less carbon emission, etc, the paper seems to make a good contribution considering the models are growing in size.

Considering that the paper demonstrates a solid idea to show good improvement in performance to solve real problem, I want to stay positive about the work.
However, the work seems to be missing out on the details and insights that led to the design, hence the score.

I would like to revise my review after the rebuttal.

**Strengths:**

+ Better utilization of pipeline parallelism of DNN training can lead to large benefits such as less cost for training, less carbon emission, etc. So the paper seems to make a good contribution considering the models are growing in size.

**Weaknesses:**

- Seems the related works seem to be missing on some key works that may paint a bigger picture of the area.
- Details seem rather light as to what led to the proposed design. For example there are works like Decima that aims to parse the computational graphs using GCNs and use RL to perform scheduling for data clusters. The problem may be different. however, it would be nice to know what "insights" led to the proposed design.

**Questions:**

* Can this be generalized into heterogeneous devices? Lets say multi GPUs and multi TPUs. What might be the technical challenges?
* Minor: Can you describe the relation of this work to Alpa presented in OSDI 2022? https://www.usenix.org/conference/osdi22/presentation/zheng-lianmin
* The overall goal of parsing a graph and deploying a task may be viewed very similar to the Decima presented in SIGCOMM 2019 https://web.mit.edu/decima/content/sigcomm-2019.pdf
Can you contrast the work in terms of what led to the design decisions that may differentiate the work from Decima? I understand that the problem that it tackles is different. However, it would be a good addition to the work if some "insights" behind the design are better outlined.
* Can you provide how the work may perform for some LLMs? I believe the computer vision networks that are orders of magnitude smaller than LLMs. Additional results for LLMs may help better understand the details. For example, in the context of how the DRL-PP performs for memory-bound vs. compute-bound task.

---

### Meta-Review · Area_Chair_fQ5d · 2023-12-09

**Metareview:**

Four knowledgeable reviewers reviewed this submission. They raised concerns w.r.t. (1) the positioning of the proposed approach w.r.t previous work (dWCJ, 3JAq, VkLQ) - they found missing key works in the area and important comparisons-, (2) the design choices which did not appeared well justified (dWCJ), (3) the unclear significance of the proposed approach (nNSt), and (4) the rather unconvincing experimental evidence (3JAq, nNSt, VkLQ). Unfortunately, there was no rebuttal. The AC agrees with the reviewers' assessment and therefore recommends to reject. The AC encourages the authors to consider the feedback provided by the reviewers to improve future iterations of their work.

**Justification For Why Not Higher Score:**

Major concerns raised by the reviewers and no rebuttal to address them.

**Justification For Why Not Lower Score:**

N/A

---

### Decision · Program_Chairs · 2024-01-16

Reject